# Precipitation during Quenching in 2A97 Aluminum Alloy and the Influences from Grain Structure

**DOI:** 10.3390/ma14112802

**Published:** 2021-05-25

**Authors:** Xiaoya Wang, Jiantang Jiang, Guoai Li, Wenzhu Shao, Liang Zhen

**Affiliations:** 1School of Materials Science and Engineering, Harbin Institute of Technology, Harbin 150001, China; wangxiaoya0316@126.com (X.W.); wzshao@hit.edu.cn (W.S.); 2National Key Laboratory of Precision Hot Processing of Metals, Harbin Institute of Technology, Harbin 150001, China; 3AVCC Beijing Institute of Aeronautical Materials, Beijing 100095, China; LLLDK3@163.com

**Keywords:** 2A97 Al–Cu–Li alloy, quench precipitation, cooling rate, recrystallized grain, sub-grains

## Abstract

The quench-induced precipitation and subsequent aging response in 2A97 aluminum alloy was investigated based on the systematic microstructure characterization. Specifically, the influence on precipitation from grain structure was examined. The results indicated the evident influence from the cooling rate of the quenching process. Precipitation of T_1_ and δ′ phase can hardly occur in the specimen exposed to water quenching while become noticeable in the case of air cooling. The yield strength of 2A97-T6 alloy de-graded by 234 MPa along with a comparable elongation when water quenching was replaced by air cooling. Sub-grains exhibited a much higher sensitivity to the precipitation during quenching. The presence of dislocations in sub-grains promoted the quench-induced precipitation by acting as nucleation sites and enhancing the diffusion of the solute. A quenching rate of 3 °C/s is tolerable for recrystallized grains in 2A97 Al alloy but is inadequate for sub-grains to inhibit precipitation. The study fosters the feasibility of alleviating quench-induced precipitation through cultivating the recrystallization structure in highly alloyed Al–Cu–Li alloys.

## 1. Introduction

Al–Cu–Li alloys have been considered as the most attractive alloys in aircraft and aerospace industries because of their high-specific strength and high-specific elastic, satisfactory corrosion resistance and low fatigue crack growth rate [1,2]. The excellent properties of these alloys are controlled by an elaborated combination of fine precipitates, including T_1_ (Al_2_CuLi), θ′ (Al_2_Cu), δ′ (Al_3_Li), β′ (Al_3_Zr), etc. [3,4]. Properly designed and executed heat treatment is basically crucial for excavating the potential of Al–Cu–Li alloys once the alloying and the forming are accomplished.

Nowadays, the demands for large-scale monolithic components are significantly increased to replace the assembly parts which need welding or riveting [5]. The need for thick plates and heavy forgings for fabricating monolithic components has then been increasing evidently. The manufacturing of hot-rolled thick plates, however, suffers from the issues related to quenching insufficiency [6]. A rapid cooling following the solution treatment is necessary to establish a supersaturated solution and obtain a high density of vacancies in the matrix to fully exploit the age-hardening potential of the alloys [7]. However, an adequate cooling rate can hardly be achieved in huge plates thoroughly due to the presence of a non-transient cooling procedure combined with the considering of internal stresses control [8,9]. As reported by Robinson [10], even when aggressive techniques like cold water spraying are used, precipitation can be unavoidable during the quenching in the core of heavy sections, which degrades the service properties including the toughness, ductility or corrosion resistance [11,12,13,14,15,16]. Deschamps [17] investigated the quench-induced precipitation of an Al–Zn–Mg–Cu alloy and found that the air cooling prior to artificial aging led to a 41% reduction in hardness, relative to that obtained in the case of water quenching. Gable [2] researched the quench sensitivity of AF/C 458 alloy under various cooling rates and found that the slow cooling resulted in decreased strength and ductility in the artificially aged alloy.

The precipitation during quenching was closely associated with the precipitation at grain boundaries and sub-grain boundaries due to their high interfacial energy [11,15,18,19,20]. These boundaries not only facilitate the formation of dispersed phase during casting [21,22,23], but are also favorable to heterogeneous nucleation during inadequate quenching. Apart from the effects from the boundaries, quenching induced precipitation can vary with grain structure. Recently, Zhang et al. [18] studied the quench-induced precipitation in 7xxx alloy and found that an appreciable amount of plate-like phase, named Y phase by the authors, precipitated only in sub-grains rather than in recrystallized grains under a wide cooling rate range of 0.05–100 °C/s. Particularly, Y phase was demonstrated isostructural to those of T_1_ phase in Al–Cu–Li alloys and the formation mechanism was considered to nucleate heterogeneously at the high-density dislocations in the sub-grains. The precipitation of Y phase individually in sub-grains was also observed by the subsequent research [19]. Actually, the plate-like T_1_ phase, identified as the most important phase in Al–Cu–Li alloys, are hard to nucleate homogeneously [24] and prefer to precipitate on grain boundaries (both low- and high-angle), dislocation loops and helices [25]. It can thus be reasonably speculated that the precipitation of T_1_ phase can be influenced by grain structure, particularly those possessing different dislocation configuration. Gable [2] found that the un-recrystallized AF/C 458 alloy was notably stronger than the recrystallized alloy in the T4 temper. The difference in strength was attributed to the Hall–Petch strengthening associated with the fine substructure, while the possible contribution from disparate precipitation between different structure was not involved in the study [2]. To our knowledge, research on the correlation between the quench-induced precipitation and grain structure in Al–Cu–Li alloys is quite limited. Therefore, a well understanding of the influence of grain structure on the precipitation during quenching will be concerned in the current work.

Moreover, the existing researches on quenching of Al–Cu–Li alloys are focused on the alloys with low solutes contents, such as 8090 alloy and AF/C 458 alloy [2,26]. The highly-alloyed Al–Cu–Li alloys deserve more attention since they are more liable to precipitation during quenching. In this work, a highly-alloyed Al–Cu–Li alloy, 2A97 Al alloy, which was newly developed in China, is selected to investigate the precipitation during quenching. The research is aimed to explore the quench-induced precipitation and subsequent aging response of 2A97 Al alloy and the influences from grain structure will be taken into consideration.

## 2. Experimental

The 2A97 Al alloy hot-rolled plate, 80 mm in thickness, was applied as starting material in the current research. The chemical composition of the alloy was shown in Table 1. Figure 1a revealed that the plate had partially recrystallized grain morphology. The un-recrystallized structure consisted of equiaxed sub-grains with an average size of 5 μm. A high density of dislocations was observed within the sub-grains according to the TEM observation shown in Figure 1c. By contrast, the recrystallized grains were free of substructure and apparently had a lower number density of dislocations, as shown in Figure 1d. There were a small number of tangled dislocations distributing in the vicinity of large Mn-containing particles in the recrystallized grains.

Samples of 10 × 10 × 10 mm^3^ in dimension were cut from the 2A97 Al alloy plate. These samples were solution treated at 520 °C for 1 h in a salt bath furnace before subjected to water quenching, air cooling or furnace cooling, and the specimens were named as WQ, AC and FC, correspondingly. To measure the temperature along with the quenching, the sample was joined firmly to a K-type thermocouple which was connected to the temperature measurement module. The temperature of the specimen was recorded by a computer that was connected to the temperature measurement module. It took 6 s and 147 s for the sample to cool from 520 °C down to 150 °C when subjected to water quenching and air quenching, presenting a cooling rate of 62 °C/s and 3 °C/s, respectively. The furnace cooling from 520 °C to 200 °C cost 16 h and the cooling rate was about 0.05 °C/s. Immediately after the cooling process, the specimens were isothermally aged at 155 °C for 55 h to T6 condition. The water-quenched, air-cooled and furnace-cooled specimens followed with aging treatments were termed as WQA, ACA and FCA, respectively.

A Differential Scanning Calorimetry (DSC) was conducted on a TA Q2000 apparatus (TA Instruments Inc., New Castle, DE, USA) in a high purity nitrogen flux. Disc samples used for DSC analysis, 3.5 mm in diameter and 0.5 mm in thickness, 20–30 mg in weight, were cut from the plate and ground thoroughly before putting into DSC analysis. The sample was heated from 40 °C to 525 °C at the ramp rate of 20 °C/min to trace the appearance of endothermic and exothermic peaks. All heat flows considered here were normalized by the sample mass.

Transmission electron microscopy (TEM) observation was carried out on an FEI Talos F200A microscope (Waltham, MA, USA) with 200 keV accelerating voltage. The TEM samples were prepared by mechanical grinding followed by electro-polishing in a twin-jet apparatus, using an electrolyte consisting of 70% methanol and 30% nitric acid at a temperature of below −20 °C at about 10 V. TEM micrographs of WQ and AC specimens were taken approximately 12 h after quenching to avoid the effect from the natural aging.

Electron backscattered diffraction (EBSD) analysis was carried out on a SUPPA 55 SAPPHIRE for scanning electron microscopy (SEM) equipped with an HKL EBSD acquisition system. Specimens EBSD were electro-polished under 20–30 V for 15 s in a solution of perchloric acid and alcohol with the ratio of 1:9. A step of 1 μm was employed for data gathering in the EBSD analysis. The EBSD data were analyzed using HKL Channel 5 system. Here in the current research grain boundaries with an angle of over 15° were certified as high angle grain boundary (HAGB) and low angle grain boundary (LAGB) of 2–15°. The HAGB and LAGB were highlighted with a black and a red line in the EBSD maps, respectively.

Samples for tensile testing were machined along the rolling direction of the plate and ground carefully before putting it into the test. The cross-section of the samples was 6 × 1.5 mm^2^ and the gauge length was 18 mm. Tensile tests were carried out at a strain rate of 1 mm/min on an Instron 5569 tensile machine. The strain was monitored by a 10 mm clip gauge extensometer.

## 3. Results

### 3.1. DSC Thermograms of Cooled Specimens

Figure 2 shows the normalized DSC thermograms for the specimens cooled by various rates from solution treatment temperature. There are three endothermic peaks (A, D and F) and three exotherm peaks (B, C and E) existing in the DSC thermogram of the WQ specimen. Endothermic peak A, observed in the temperature range of 90–120 °C, marked the dissolution of Li clusters and GP zones, referring to the previous research [27,28,29]. The exotherm peak B at approximately 138 °C indicates the precipitation of GP zones [27,28,30,31,32]. The precipitation of δ′ phase contributes to the peak C at about 180 °C [27,28,30,31,32]. The appearance of the endothermic peak D between 190 °C and 230 °C proves the dissolution of GP zones and δ′ phase that formed in the temperature range of peak B and peak C [27,28,30,31,32]. The intense exotherm peak, marked by peak E, is observed in the temperature range of 250–320 °C. Despite the coexistence of T_1_, δ, S′ and T2 phase reported by the literature [28,30,31,33], the precipitation of the T_1_ phase is believed to contribute dominantly to the exotherm [34,35,36]. The endothermic peak F is observed in the temperature range of 370–500 °C, which is attributed to the dissolution of the T_1_ phase and other high-temperature phases.

Compared to the thermal response of the WQ specimen, the DSC curve of the AC specimen presents significantly different characteristics. The first endothermic peak of the AC specimen, labelled as peak A′, occurs between 120 °C and 190 °C. This peak is believed to rise from the dissolution of GP zones and clusters [37,38]. Compared to that of the WQ specimen, the lower cooling rate of AC treatment results in the coarsening of GP zones and thus higher thermal stability. In such cases, peak A shifts towards higher temperatures to form peak A′. Besides, the larger area of peak A′ comparing to peak A suggests the increased amount of GP zones in the AC specimen. Similarly, the endothermic peak D shifts towards higher temperature together with an increased peak area, suggesting the sufficient precipitation of δ′ phase during the AC treatment. Moreover, the exotherm peak E′ caused by precipitation of the T_1_ phase is also observed at a temperature higher than that in the case of WQ, suggesting that the precipitation kinetics of the T_1_ phase is reduced in the AC specimen. This is consistent with the previous observations that the onset of T_1_ precipitation is delayed due to the slower cooling rate [39]. Besides, the peak area of E′ is significantly smaller than that of peak E, suggesting that the driving force for the precipitation of the T_1_ phase is decreased in the AC specimen. Whereas the peak area of the endothermic peak F′, which is related to the dissolution of the T_1_ phase, is evidently larger than that of peak F in the case of water quenching. In terms of peak E′ and peak F′, it is reasonably speculated that there is significant precipitation of the T_1_ phase that occurred during the AC treatment. As to the FC specimen, the DSC thermogram only shows a quite small exotherm peak (peak E′′) that corresponds to the precipitation of the T_1_ phase and a small endothermic peak (peak F′′) related to the dissolution of the T_1_ phase. The precipitation of strengthening phases mostly disappears in the DSC trace of the FC specimen. It is suggested that very strong precipitation may occur during the FC process. The insufficient dissolution of the T_1_ phase at peak F′′ may be related to the large-sized T_1_ phase formed in the FC specimen.

In addition, the precipitation of GP zones and δ′ phase that are distinguished (peak B and peak C) in the WQ specimen, can hardly be recognized in the AC and FC specimen. A similar phenomenon is previously observed in Chen’s research [23], which noted that the precipitation of GP zones and δ′ phase was drastically reduced when the solution treated specimens was cooled at slow rates

### 3.2. Precipitation in Recrystallized Grains and Sub-Grains

The TEM images of 2A97 aluminum alloy solution treated at 520 °C and cooled in various ways were shown in Figure 3. In the WQ specimen, only a few fine spherical particles are existing in the recrystallized grains as well as sub-grains, as shown in Figure 3a,b. The weak spot located at sites of {001}Al in the selected area electron diffraction (SAED) pattern corresponding to <011>Al reveals the presence of the L12 phase. The spherical particles are regarded to be δ′ precipitates, which are previously observed in water-quenched Al–Cu–Li alloys [2,26,40] due to their small interfacial energy and nucleation barrier [41]. When the alloy is exposed to AC treatment, the diffraction spots belonging to the Al matrix and δ′ phases can be distinguished in the SAED pattern taken from the recrystallized grain, as shown in Figure 3c. A few plate-like precipitates are observed in the recrystallized grain, as marked by a circle in Figure 3c. These fine plates are identified to be T_1_ phase on the basis of the characteristic orientation that parallels with {111} planes of the Al matrix. The T_1_ precipitates in the recrystallized grains have an average diameter of 122 nm, which is comparable to that observed in Al–Li alloys of the peak-aged condition [3,4]. By contrast, the SAED taken from the region of sub-grains reveals three sorts of diffraction patterns related to the matrix, T_1_ phases and δ′ phases, as shown in Figure 3d. A considerable number of T_1_ phase is observed in the sub-grains of the AC specimen. Notably, these T_1_ precipitates have a significantly larger number density than those observed in recrystallized grains. What’s more, the precipitates of the T_1_ phase in sub-grains are 208 nm averagely in diameter, which are apparently larger than those formed in recrystallized grains. The observations suggest that the precipitation of the T_1_ phase in sub-grains is more sufficient than that in recrystallized grains. When the alloy is exposed to furnace cooling, the difference between the precipitation in recrystallized grains and that in sub-grains becomes more striking according to Figure 3e,f. The SAED pattern of the recrystallized grains presents strong diffraction belonging to the T_1_ phase [3]. A large number of plate-like T_1_ phases are observed in the recrystallized grains and measured 0.9 μm averagely in diameter. In comparison, there are plenty of large plate-like precipitates existing in the un-recrystallized grain. These large precipitates coarsen up drastically to 5.7 μm averagely in diameter and mostly run through several sub-grains. It will be shown subsequently that these large plates are T_1_ phase.

Figure 4 shows the further observations on the large plate-like precipitates in the un-recrystallized grain of the FC specimen. The high-resolution transmission electron microscopy (HRTEM) of a single plate shows that the (0001) plane of the phase exhibit a spacing of 0.47 nm and is paralleled to the (111) plane of the matrix, which agrees well with the widely recognized characteristic of the T_1_ phase [42]. The T_1_ phase was described as a hexagonal lattice on {111}Al and has an orientation relationship of [0001]T_1_//[111]α and [10–10]T_1_//[110]α [39]. The inset fast Fourier transform (FFT) reveals the diffraction patterns including parts of the plate and matrix, confirming the presence of the T_1_ phase. These T_1_ precipitates grow up more preferentially along the length but less along the short axis and the thickness, resulting in a high aspect ratio.

To further investigate the correlation between the precipitation of the T_1_ phase and the local grain structure, the FC specimen is examined by EBSD and the representative results are shown in Figure 5. The needle-like etch grooves, as shown in the SE image in Figure 5a, evolves from the dissolution of the T_1_ phase during electro-polishing, and thus demonstrates the pre-existence of the T_1_ phase. It is found that the T_1_ phase has significantly different sizes among grains. The grains containing a larger-sized T_1_ phase are outlined with a black dash line, as shown in Figure 5b. The grain orientation spread (GOS) map, which provides a measure of intra-granular lattice distortion, is used to identify recrystallized grains. Generally, the grain with an orientation spread near 6o is regarded as a fully recovered grain, while a fully recrystallized grain is expected to exhibit a much lower orientation spread [43,44,45]. A threshold of GOS ≤3 is adopted herein to define the cut-off value for recrystallized grains referring to previous research [46]. The GOS map reveals the distribution of re-crystallized grains in Figure 5c. It can be observed that the recrystallized grains are consistent exactly with the grains containing fine T_1_ phase, while the un-recrystallized grains accord with the grains containing larger-sized T_1_ phase in Figure 5b. Moreover, the un-recrystallized grains possess a good deal of sub-grains according to the grain boundary (GB) shown in Figure 5d. In contrast, few low-angle grain boundaries can be recognized in the recrystallized grains. The EBSD observations suggest that the T_1_ precipitates coarsen up more significantly in the un-recrystallized grains than in recrystallized grains.

Figure 6 shows the phases formed at grain boundary and sub-boundary during the air cooling and furnace cooling process. There are a few plate-like phases existing at sub-boundaries in the AC specimen. These phases are identified to be T_1_ phase via the HRTEM observing and FFT analysis, as shown in Figure 6b. These T_1_ precipitates are 144 nm averagely in diameter, which are relatively larger than those formed in 2A97 alloy of peak-aged conditions [34,35]. The grain boundaries in the AC specimen are found to be free of precipitation according to the TEM observation in Figure 6c. Moreover, discontinuously distributed precipitates companied by a precipitation-free zone of 2 μm in width are found developed at grain boundaries in the FC specimen, as shown in Figure 6d.

### 3.3. Effects of Cooling Rates on Tensile Properties

Tensile tests are performed to quantify the artificial aging response as a function of the cooling rate. The WQ, AC and FC specimens are artificially aged at 155 °C for 55 h to approximately T6 condition before subjected to the tensile tests. As shown in Figure 7, the WQA specimen exhibits yield strength (YS) of 566 MPa and ultimate tensile strength (UTS) of 613 MPa, which is comparable to that of peak-aged 2A97 alloy reported previously [34,35]. The WQA specimen has an elongation of 8.5%, indicating a good match of strength and plasticity. The ACA specimen presents an elongation comparable to the WQA specimen, while the YS and UTS are degraded by 234 MPa and 156 MPa, respectively. The alloy is thus considered relatively quench sensitive in terms of T6 strength. The loss of aging hardening is closely related to the significant precipitation during AC treatment, particularly that occurs in sub-grains. The FCA specimen exhibits the lowest overall strength and poor elongation and this can be adequately explained by the inappreciable precipitation events in the DSC trace.

## 4. Discussion

When the alloy is exposed to AC treatment, the T_1_ phase in sub-grains has significantly larger sizes and number density than that in recrystallized grains. Moreover, the sizes of the T_1_ phase in sub-grains of AC specimen are even larger than that in over-aged Al–Cu–Li alloys [4,47]. The different precipitation of the T_1_ phase between recrystallized grains and sub-grains is further striking when the alloy is exposed to FC treatment. Although furnace cooling is exaggeratively slow for practical application, it reveals the tendency that the disparate T_1_ precipitation will be more remarkable with a further decrease in cooling rate. To further investigate the precipitation of the T_1_ phase, the FC treatment is interrupted at 420 °C to observe the precipitation at the early stage, and the TEM observations are shown in Figure 8. A few fine T_1_ phase is observed in the sub-grain according to Figure 8a. Notably, these fine T_1_ particles are found sprouted from dislocations. T_1_ precipitates of around 20 nm in diameter are also found at sub-boundary, as shown in Figure 8b. It is suggested that the precipitation of the T_1_ phase depends closely on dislocations and sub-boundaries. Meanwhile, there are hardly any T_1_ phase recognized within the recrystallized grains at the early stage of FC treatment. The disparity in the precipitation of the T_1_ phase, in sub-grains and recrystallized grains, is dictated by the density of heterogeneous nucleation sites. Cassada [48] observed that the nucleation of T_1_ plates occurs at dislocations partials where the growth interfaces and edges evolve gradually. This is consistent with the mechanism proposed by Noble and Thompson [39]. As such, a small strain prior to the artificial aging is routinely used to introduce a uniform distribution of dislocations in the matrix, which serve as nucleation sites for the T_1_ phase [3]. Besides, the quick diffusion along dislocations and sub-boundaries is contributive for promoting the nucleation and growth of the T_1_ phase [49].

The precipitation in 2A97 Al alloy is effectively prevented by water quenching (62 °C/s) from solution treatment temperature. However, the precipitation of the T_1_ phase and δ′ phase occurs sufficiently during the air cooling (3 °C/s) and results in significantly degraded aging hardening. The alloy is thus considered sensitive to cooling rate according to the strength comparison in the T6 condition. As reported by Chen [26], 8090 Al–Li alloy was regarded to possess low quench sensitivity since the T6 strength reduced only by 35 MPa in air-cooling conditions. Work by Gable [2] illustrated that the yield and tensile strength of AF/C 458 Al–Li alloy was comparable in the case of water quenching, glycerol quenching and air cooling, and thus the alloy was considered relatively quench insensitive. The low quench sensitivity of these alloys was related to the modest number of solute elements [50]. In comparison, the highly alloyed solutes contribute to the excellent performances of 2A97 alloy in density, specific strength, elastic modulus, etc., while it meanwhile leads to noticeable precipitation in the case of inadequate quenching process. The copper level especially promoted the quench-induced precipitation according to the research by Shakesheff [40].

As for the 10 mm ×10 mm × 10 mm 2A97 alloy specimen, a cooling rate of 3 °C/s (air cooling) is found inadequate to prohibit the quenching precipitation and subsequent performance degradation. In terms of production, this means that 2A97 alloy has a relatively limited quenching window if the desired mechanical property is the strength or the production has a huge sectional dimension. Considering the influences from grain structure, it is demonstrated that the recrystallized grains exhibit significantly lower sensitivity to the precipitation during quenching. Specifically, the recrystallized grains of 2A97 alloy can tolerate a cooling rate of 3 °C/s, or even slower. This fosters the feasibility of alleviating the quench-induced precipitation by cultivating recrystallization, which is favorable for the mechanical properties of large-scale components. Moreover, it is acknowledged that recrystallized microstructure can significantly reduce the in-plane and through-thickness anisotropy of thick plates. Therefore, further researches on recrystallization are believed to rise in larger-scale monolithic components of Al–Cu–Li alloys.

## 5. Conclusions

The quench-induced precipitation and subsequent aging response were investigated under various cooling rates (water quenching, air cooling and furnace cooling) applied after solution treatment. The precipitation in recrystallized grains and un-recrystallized grains consisting of the substructure was characterized. Conclusions obtained from the present work are drawn:The influence from the cooling rate during the quenching is evident in 2A97 Al alloys. Precipitation can hardly occur during the water quenching, while the precipitation of T_1_ and δ′ phase was detected in the air-cooled specimen;Quenching precipitation leads to degraded mechanical properties. In comparing the ACA specimen with the WQA specimen, the YS of 2A97 alloy is degraded by 234 MPa and UTS by 156 MPa;Sub-grains exhibited much higher sensitivity to quenching precipitation. The presence of dislocations in sub-grains promoted quenching precipitation by acting as nucleation sites and enhancing solutes diffusion;The quenching rate of 3 °C/s is tolerable for the recrystallized grains in 2A97 Al alloy but is inadequate for sub-grains to prohibit precipitation.

## Figures and Tables

**Figure 1 materials-14-02802-f001:**
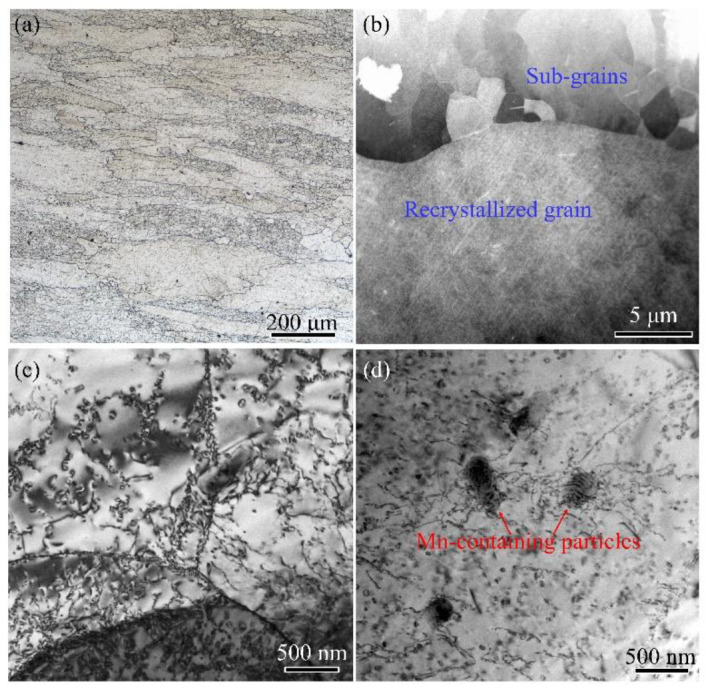
Metallograph and TEM images of the 2A97 Al alloy plate: (**a**) metallograph image; (**b**) TEM image; (**c**) dislocation in sub-grains; (**d**) dislocation in recrystallized grain.

**Figure 2 materials-14-02802-f002:**
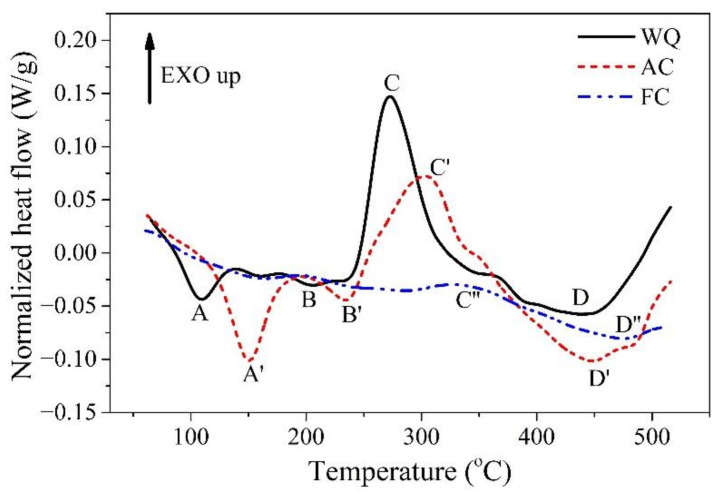
Normalized heat flux of the 2A97 Al alloy solution treated at 520 °C for 1 h and cooled by different methods.

**Figure 3 materials-14-02802-f003:**
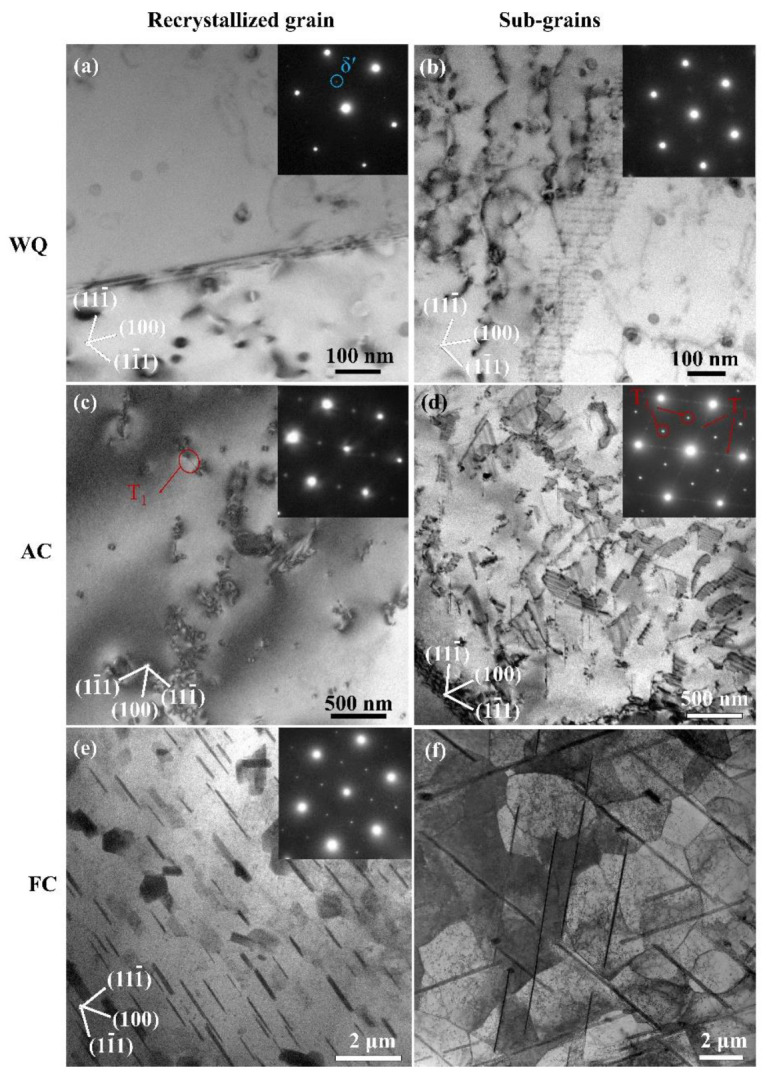
TEM images of the precipitates in recrystallized and sub-grains in 2A97 alloy that scheme 520 °C for 1 h and cooled by WQ, AC and FC: precipitates in recrystallized grain of (**a**) WQ, (**c**) AC and (**e**) FC; precipitates in sub-grains of (**b**) WQ, (**d**) AC and (**f**) FC.

**Figure 4 materials-14-02802-f004:**
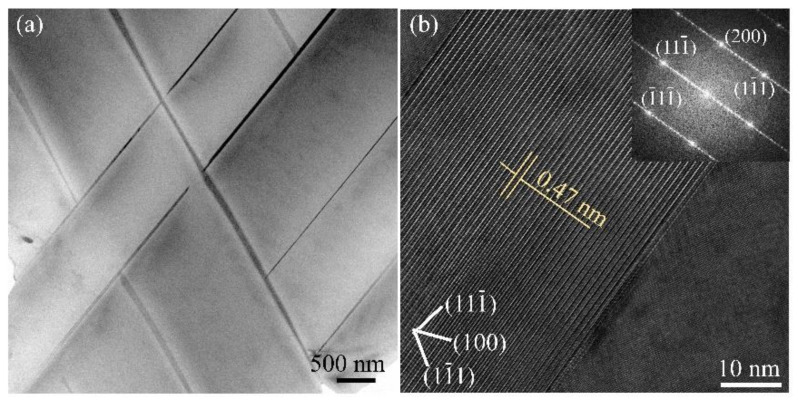
Large T_1_ phase in the un-recrystallized grains in 2A97 alloy that solution treated at 520 °C for 1 h and cooled by FC: (**a**) TEM image; (**b**) HRTEM image and FFT pattern of a single plate-like phase in (**a**).

**Figure 5 materials-14-02802-f005:**
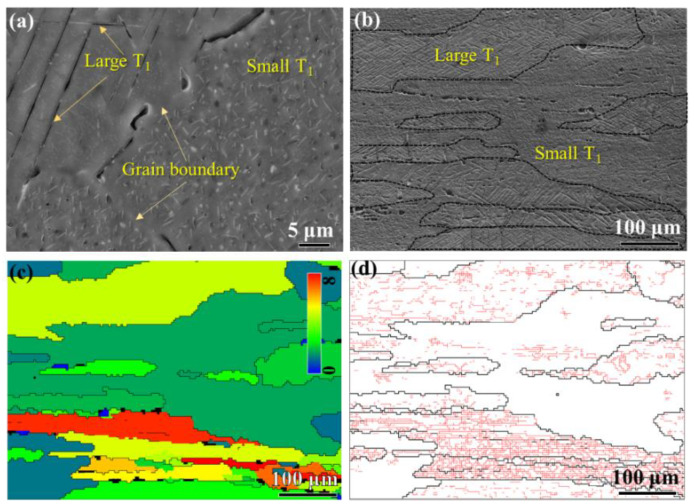
Correlation between the precipitation of T_1_ phase and the local grain structure in 2A97 alloy that solution treated at 520 °C for 1 h and cooled by FC: (**a**) SE image at high magnification; (**b**) SE image at low magnification; (**c**) GOS map and (**d**) GB map corresponding to (**b**).

**Figure 6 materials-14-02802-f006:**
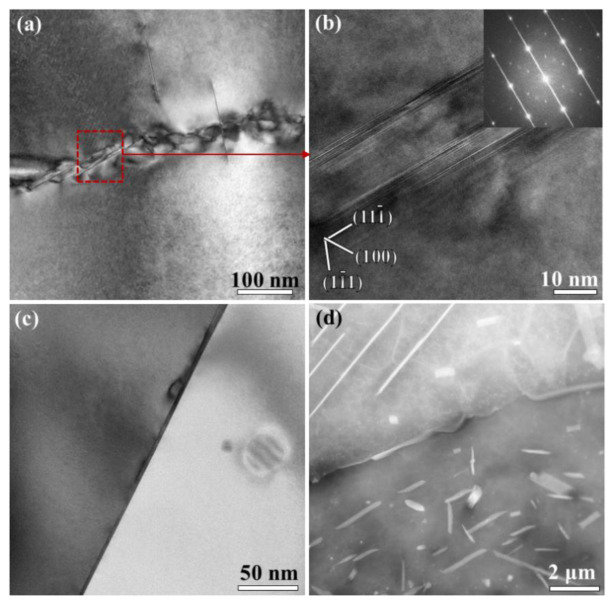
TEM images of the precipitates at sub-boundary and grain boundary in 2A97 alloy that solution treated at 520 °C for 1 h and cooled by AC and FC: (**a**) sub-boundary of AC; (**b**) HRTEM and FFT of the boxed region in (**a**); (**c**) grain boundary of AC; (**d**) grain boundary of FC.

**Figure 7 materials-14-02802-f007:**
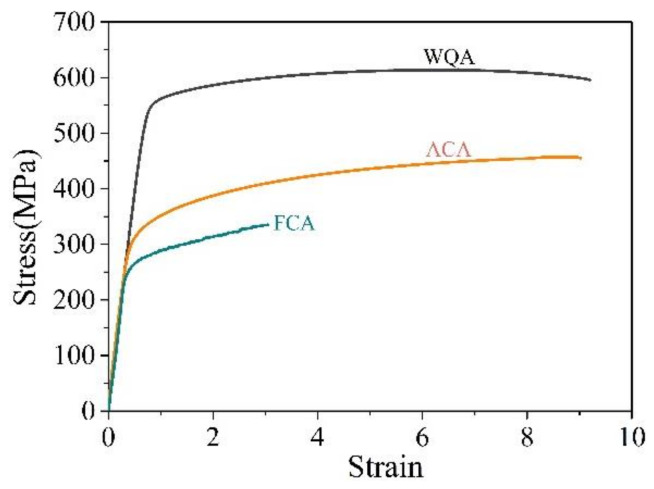
Tensile properties of 2A97 Al alloy that solution treated at 520 °C for 1 h, cooled by various rates and aged at 155 °C for 55 h.

**Figure 8 materials-14-02802-f008:**
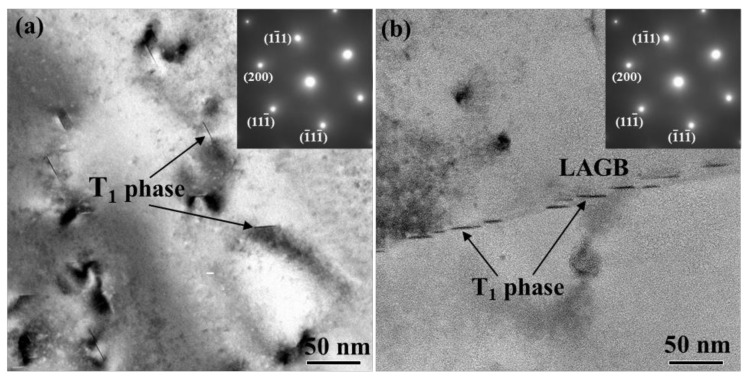
TEM images of the precipitates in 2A97 alloy that solution treated at 520 °C for 1 h and cooled down to 420 °C by FC: (**a**) sub-grains interior; (**b**) low angle grain boundary.

**Table 1 materials-14-02802-t001:** Composition of the 2A97 alloy under investigation (wt%).

Cu	Li	Mg	Zn	Mn	Zr	Al
3.55	1.40	0.44	0.46	0.29	0.11	Bal.

## Data Availability

The data presented in this study are available on request from the corresponding author.

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
