# Peer review of "Precipitation during Quenching in 2A97 Aluminum Alloy and the Influences from Grain Structure"

_materials, 2021, doi:10.3390/ma14112802_

Round 1
Reviewer 1 Report
The paper shows the research that is aimed to explore the quench precipitation and subsequent ageing response of 2A97 alloy.
The results show that in 2A97 alloy, precipitation can hardly occur during the water quenching, while the precipitation of T1 and δ′ phase was detected in the air cooled specimen. Precipitation during the quenching leads to degraded mechanical properties. Subgrains exhibited much higher sensitivity to precipitation during the quenching. The cooling rate of higher than 3 °C/s is necessary for the unrecrystallized grains containing substructure while recrystallized grains can tolerate a lower cooling rate.
The work is interesting but should be improved. Manuscript can be accepted for publication after minor revision.

Author Response
Reply to the reviewers’ comments-1
Thanks a lot for the reviewers’ comments on the manuscript that we submitted to the journal of Materials (No. Materials_1155513). The authors have revised the manuscript carefully regarding on the comments.
As the comments was made with label on the PDF, the comments are summarized as following:
- There are a lot of format problems such as the symbol of unit, dashes in a word, reference.
Response: Thanks very much for the suggestion. The text was arranged carefully according to the template before submitting. The manuscript in Word version was all right but the PDF version were messed up. The author will contact with the editorial office for help and handle these issues.
- The abbreviation could be defined on first use.
Response: Thanks for the reminder. All the abbreviations have been defined on the first use in the revised manuscript.
- What was the dimension of the samples, and if samples are too big, where the cooling time were measured.
Response: Thanks very much for the valuable comment! The samples are 10mm×10mm×10mm in dimension. The dimension of the specimens and the measurement of temperature has been modified in the revised manuscript.
The corresponding revisions are as follows—
“The samples of 10mm×10mm×10mm in dimension were cut from the 2A97 Al alloy plate. These samples were solution treated at 520 oC for 1 h in a salt bath furnace before subjected to water quenching, air cooling or furnace cooling and the specimens were named as WQ, AC and FC correspondingly.”
“To measure the temperature all along the quenching, the sample was joined firmly to a K-type thermocouple which was connected to the temperature measurement module. The temperature was recorded by a computer that was connected to the temperature measurement module. It took 6 s and 147 s for the samples to cool from 520 oC down to 150 oC when subjected to water quenching and air quenching, presenting a cooling rate of 62 oC/s and 3 oC/s respectively. The furnace cooling from 520 oC to 200 oC cost 16 h and the cooling rate was about 0.05 oC/s.”
- The fourth conclusion “The cooling rate of higher than 3oC/s is necessary for the un-recrystallized grains containing sub-structure while recrystallized grains can tolerate a lower cooling rate.” is little bit confusing.
Response: Thanks for the suggestion. This sentence has been rearranged as “The quenching rate of 3 oC/s is tolerable for the recrystallized grains in 2A97 Al alloy but is inadequate for sub-grains to prohibit precipitation.”

Reviewer 2 Report
This document is a good piece of research, although we can find room to improvement:
- Authors present a lot of hyphenisation words in middle of phrases, such as "ad-equate" line 39 , but also in line 41, 42, 51, 74, 101, 116, 133, 136, and much more, so a careful reading is recommended;
- An excess text, not related to the research topic is on the paper, between line 78 to line 93;
- Line 115, 120, 121 and 122 oC should be changed by º C.
- Caption of figure 5, line 271-273 should be in the same page as images
Author Response
Reply to the reviewers’ comments-2
Thanks a lot for the reviewers’ comments on the manuscript that we submitted to the journal of Materials (No. Materials_1155513). The authors have revised the manuscript carefully regarding on the comments.
This document is a good piece of research, although we can find room to improvement:
- Authors present a lot of hyphenation words in middle of phrases, such as "ad-equate" line 39, but also in line 41, 42, 51, 74, 101, 116, 133, 136, and much more, so a careful reading is recommended;
Response: Thanks very much for the suggestion! The text was arranged carefully according to the template before submitting. The manuscript in Word format was all right but the PDF version were messed up. The author will contact with the editorial office for help and handle these issues.
2.An excess text, not related to the research topic is on the paper, between line 78 to line 93;
Response: The excess text was not involved in the text before submitting but came out in the PDF version. The text has been deleted in the revised manuscript and this issue will be settled subsequently.
- Line 115, 120, 121 and 122 oC should be changed by º C.
Response: This has been improved in the revised manuscript.
- Caption of figure 5, line 271-273 should be in the same page as images
Response: Thanks for the suggestion! The text has been rearranged to make sure the figure 5 and related caption presenting in the same page.

Reviewer 3 Report
It is a good paper devoted to the very important material, namely the 2A97 aluminum alloy. Al–Cu–Li alloys have been considered as the most attractive alloys in aircraft and aerospace industries because of their high-specific strength and high-specific elastic, satisfactory corrosion resistance and low fatigue crack growth rate. The paper could be published after major revisions. In particular, it is visible in the micrographs that the grain boundaries of the Al-based matrix can contain the continous layers of the precipitates (see Fig 6d) if these grain boundaries are high-angle and, therefore, have high energy. To the contrary, the low-angle grain boundaries (having low energy) contain the interrupted chain of the precipitates (see Fig 8b). Such behaviour is intimately connected with the so-called complete and incomplete wetting of high- and low-angle grain boundaries by the melt or second solid phase (see for example JETP Letters 88 (2008) 537 or Metals 10 (2020) 1127). The morphology of integranular phase strongly influences the overall properties of a polycrystalline composite (see for example Mater. Sci. Forum 633 (2009) 321). The grain boundary wetting transitions can definitely influence the phenomena observed by the authors. I would strongly propose to discuss these points in the paper.
Author Response
Reply to the reviewers’ comments-3
Thanks a lot for the reviewers’ comments on the manuscript that we submitted to the journal of Materials (No. Materials_1155513). The authors have revised the manuscript carefully regarding on the comments.
It is a good paper devoted to the very important material, namely the 2A97 aluminum alloy. Al–Cu–Li alloys have been considered as the most attractive alloys in aircraft and aerospace industries because of their high-specific strength and high-specific elastic, satisfactory corrosion resistance and low fatigue crack growth rate. The paper could be published after major revisions. In particular, it is visible in the micrographs that the grain boundaries of the Al-based matrix can contain the continous layers of the precipitates (see Fig 6d) if these grain boundaries are high-angle and, therefore, have high energy. To the contrary, the low-angle grain boundaries (having low energy) contain the interrupted chain of the precipitates (see Fig 8b). Such behaviour is intimately connected with the so-called complete and incomplete wetting of high- and low-angle grain boundaries by the melt or second solid phase (see for example JETP Letters 88 (2008) 537 or Metals 10 (2020) 1127). The morphology of integranular phase strongly influences the overall properties of a polycrystalline composite (see for example Mater. Sci. Forum 633 (2009) 321). The grain boundary wetting transitions can definitely influence the phenomena observed by the authors. I would strongly propose to discuss these points in the paper.
Response: Thanks very much for the valuable suggestion! There are visible precipitates at grain boundary in Fig.6d and these particles forms during melting or casting process. This content has been added in the Introduction in the revised manuscript. Besides, the precipitation of plate-like phases at low-angle grain boundary during inadequate quenching is focused in the present work and the related influences on properties of the alloy are discussed in detail.
The corresponding revisions are as follows—
“The precipitation during quenching was closely associated with the precipitation at grain boundaries and sub-grain boundaries due to their high interfacial energy [R1-R5]. These boundaries not only facilitate the formation of dispersed phase during casting [R6-R8], but also are favorable to heterogeneous nucleation during quenching with lower rate and ageing treatment.”
Reference
R1. Zhang, Y.; Weyland, M.; Milkereit, B.; Reich, M.; Rometsch, P.A. Precipitation of a new platelet phase during the quenching of an Al-Zn-Mg-Cu alloy. Scientific Reports 2016, 6, 23109-23117, doi:10.1038/srep23109.
R2. Liu, S.; Li, Q.; Lin, H.; Sun, L.; Long, T.; Ye, L.; Deng, Y. Effect of quench-induced precipitation on microstructure and mechanical properties of 7085 aluminum alloy. Materials & Design 2017, 132, 119-128, doi:10.1016/j.matdes.2017.06.054.
R3. Godard, D.; Archambault, P.; Aeby-Gautier, E.; Lapasset, G. Precipitation sequences during quenching of the AA 7010 alloy. Acta Mater. 2002, 50, 2319-2329, doi:https://doi.org/10.1016/S1359-6454(02)00063-0.
R4. Song, F.; Zhang, X.; Liu, S.; Tan, Q.; Li, D. The effect of quench rate and overageing temper on the corrosion behaviour of AA7050. Corros. Sci. 2014, 78, 276-286, doi:10.1016/j.corsci.2013.10.010.
R5. Deschamps, A.; Texier, G.; Ringeval, S.; Delfaut-Durut, L. Influence of cooling rate on the precipitation microstructure in a medium strength Al–Zn–Mg alloy. Mater. Sci. Eng. A 2009, 501, 133-139, doi:https://doi.org/10.1016/j.msea.2008.09.067.
R6. Valiev, R.; Murashkin, M.Y.; Straumal, B.B. Enhanced Ductility in Ultrafine-Grained Al Alloys Produced by SPD Techniques. Mater. Sci. Forum 2009, 633-634, 321-332, doi:10.4028/www.scientific.net/MSF.633-634.321.
R7. Straumal, A.; Mazilkin, I.; Tzoy, K.; Straumal, B.; Bryła, K.; Baranchikov, A.; Eggeler, G. Bulk and Surface Low Temperature Phase Transitions in the Mg-Alloy EZ33A. Metals 2020, 10, 1127, doi:10.3390/met10091127.
R8. Straumal, B.B.; Bokshtein, B.S.; Straumal, A.B.; Petelin, A.L. First observation of a wetting phase transition in low-angle grain boundaries. JETP Letters 2009, 88, 537-542, doi:10.1134/s0021364008200149.

Round 2
Reviewer 3 Report
After revision the paper is acceptable for publication as it is